# Future Directions in the Assessment of Axillary Lymph Nodes in Patients with Breast Cancer

**DOI:** 10.3390/medicina59091544

**Published:** 2023-08-25

**Authors:** Filippo Pesapane, Luciano Mariano, Francesca Magnoni, Anna Rotili, Davide Pupo, Luca Nicosia, Anna Carla Bozzini, Silvia Penco, Antuono Latronico, Maria Pizzamiglio, Giovanni Corso, Enrico Cassano

**Affiliations:** 1Breast Imaging Division, IEO European Institute of Oncology IRCCS, 20141 Milan, Italy; anna.rotili@ieo.it (A.R.); luca.nicosia@ieo.it (L.N.); anna.bozzini@ieo.it (A.C.B.); silvia.penco@ieo.it (S.P.); antuono.latronico@ieo.it (A.L.); maria.pizzamiglio@ieo.it (M.P.); enrico.cassano@ieo.it (E.C.); 2Breast Imaging Division, AOU Città della Scienza e della Salute di Torino, 10126 Turin, Italy; luciano.mariano@ieo.it; 3Division of Breast Surgery, IEO European Institute of Oncology IRCCS, 20141 Milan, Italy; francesca.magnoni@ieo.it (F.M.); giovanni.corso@ieo.it (G.C.); 4European Cancer Prevention Organization (ECP), 20122 Milan, Italy; 5Radiology Division, Department of Precision Medicine, University of Campania Luigi Vanvitelli, 80138 Naples, Italy; davide.pupo@ieo.it; 6Department of Oncology and Hemato-Oncology, University of Milan, 20122 Milan, Italy

**Keywords:** lymph nodes, breast cancer, radiology, surgery, cancer, oncology

## Abstract

*Background and Objectives*: Breast cancer (BC) is a leading cause of morbidity and mortality worldwide, and accurate assessment of axillary lymph nodes (ALNs) is crucial for patient management and outcomes. We aim to summarize the current state of ALN assessment techniques in BC and provide insights into future directions. *Materials and Methods*: This review discusses various imaging techniques used for ALN evaluation, including ultrasound, computed tomography, magnetic resonance imaging, and positron emission tomography. It highlights advancements in these techniques and their potential to improve diagnostic accuracy. The review also examines landmark clinical trials that have influenced axillary management, such as the Z0011 trial and the IBCSG 23-01 trial. The role of artificial intelligence (AI), specifically deep learning algorithms, in improving ALN assessment is examined. *Results*: The review outlines the key findings of these trials, which demonstrated the feasibility of avoiding axillary lymph node dissection (ALND) in certain patient populations with low sentinel lymph node (SLN) burden. It also discusses ongoing trials, including the SOUND trial, which investigates the use of axillary ultrasound to identify patients who can safely avoid sentinel lymph node biopsy (SLNB). Furthermore, the potential of emerging techniques and the integration of AI in enhancing ALN assessment accuracy are presented. *Conclusions*: The review concludes that advancements in ALN assessment techniques have the potential to improve patient outcomes by reducing surgical complications while maintaining accurate disease staging. However, challenges such as standardization of imaging protocols and interpretation criteria need to be addressed. Future research should focus on large-scale clinical trials to validate emerging techniques and establish their efficacy and cost-effectiveness. Over-all, this review provides valuable insights into the current status and future directions of ALN assessment in BC, highlighting opportunities for improving patient care.

## 1. Introduction

BC is a significant cause of morbidity and mortality worldwide, with ALN involvement being an important prognostic factor that influences patient management and outcomes [1,2].

Radiological assessment of ALNs has become increasingly important in recent years, with the aim of providing accurate information for disease staging, treatment planning, and follow-up [3].

In recent years, there have been significant advancements in assessment techniques for ALN evaluation, which have the potential to improve patient outcomes.

This narrative review aims to summarize the current state of assessment of ALNs in patients with BC and provide insight into future directions.

### 1.1. Anatomy Pills

The American Joint Committee on Cancer has categorized regional lymph nodes as follows: 1. Axillary (ipsilateral): this category includes interpectoral (Rotter’s) nodes and nodes located along the axillary vein and its tributaries. 2. Internal mammary (ipsilateral): this category comprises nodes found in the inter-costal spaces along the edge of the sternum in the endo-thoracic fascia. 3. Supraclavicular: nodes within the supraclavicular fossa, forming a triangle defined by the omohyoid muscle and tendon (lateral and superior border), the internal jugular vein (medial border), and the clavicle and subclavian vein (lower border). 4. Intramammary: nodes situated within breast tissue (designated as ALNs for N categorization and staging purposes).

Additionally, the ALNs are classified into three levels. Level I (low-axilla) refers to lymph nodes situated lateral to the lateral border of the pectoralis minor muscle. Level II (mid-axilla) refers to lymph nodes located between the medial and lateral borders of the pectoralis minor muscle and the interpectoral (Rotter’s) lymph nodes. Level III (apical axilla) refers to lymph nodes medial to the medial margin of the pectoralis minor muscle and inferior to the clavicle, also known as infraclavicular nodes.

### 1.2. Current Techniques

Many efforts have been made to establish the best diagnostic test to assess ALN status prior to treatment, thus avoiding overtreatment of women with ALN-negative BC. Removal of the involved lymph nodes was considered crucial for cure, as was that of the uninvolved lymph nodes for adequate staging [4,5]. Although NSABP B-04 results show no improvement in survival with ALND [6], such a strategy remained necessary to identify patients with lymph node involvement for whom adjuvant chemotherapy has been indicated. Axillary management was revolutionized in the early 1990s with the turning point of SLNB [7]. SLNB has become the treatment of choice and standard practice in BC surgery for axillary staging in the clinically lymph-node-negative patients [1,8,9].

Recent decades of de-escalation studies, as well as the increasing understanding of tumor biology and multigene assays in determining adjuvant tailored therapy decisions, together with improved accuracy of imaging techniques in predicting the presurgical lymph node status, have greatly influenced axillary surgical management of early BC [8,10]. These great scientific changes are promoting a further impulse towards axillary surgical conservation, even towards an omission of SLNB in selected subgroups of BC patients.

Imaging techniques such as Ultrasound (US), Computed Tomography (CT), Magnetic Resonance Imaging (MRI), and Positron Emission Tomography (PET) have been used as adjuncts to these invasive procedures [11]. US is a non-invasive, repeatable, and widely available technique that is useful for detecting metastatic lymph nodes based on morphological and functional criteria, such as dimension, shape, size, cortical thickness, margins, presence of microcalcifications and vascularization patterns using the Color Doppler technique. Additionally, US offers the option of immediate image-guided intervention. Lymph nodes are considered suspect with a ratio between longitudinal (L) and trans-verse (T) diameters > 2, ratio between hilar region and L diameters < 50%, an eccentric cortical thickening > 2 mm, and with peripheral or mixed (hilar and peripheral) vascularization patterns [12,13,14]. Recent advances in US technology, such as the use of contrast-enhanced US and elastography, have improved the diagnostic accuracy of US for ALN assessment.

Although MRI can provide information on lymph node size, shape, signal intensity, and enhancement patterns, its use for ALN assessment is limited by its high cost, limited availability, and the need for intravenous contrast administration.

PET with computed tomography (CT) is a hybrid imaging modality that can provide both anatomical and functional information: PET-CT has been shown to be useful for detecting metastatic ALN in BC patients, based on increased glucose metabolism in cancer cells. However, PET-CT is limited by its low spatial resolution and inability to distinguish between reactive and metastatic lymph nodes.

Therefore, only axillary ultrasound (AUS), if necessarily followed by US-FNAC, is used routinely in clinical practice [1]. Nevertheless, some authors are skeptical about the sensitivity and the negative predictive value (NPV) of AUS and FNAC, which vary widely according to the underlying prevalence of ALN positivity in the studied population [15,16].

### 1.3. Extracapsular Infiltration of Metastatic Lesions in Axillary Sentinel Nodes

Extracapsular infiltration refers to the extension of tumor cells beyond the lymph node capsule into the surrounding perinodal tissue. This phenomenon signifies a more advanced stage of nodal involvement and is associated with increased tumor burden, higher rates of recurrence, and reduced overall survival rates [17]. Fujii et al. [17] performed a biopsy of the SLN on 276 patients diagnosed with BC who had clinically negative ALNs and found that all cases of positive nodes in non-SLN had extracapsular infiltration at the metastatic SLNs. Their results suggested that the presence of extracapsular infiltration at metastatic SLNs is a strong predictor for residual disease in the axilla. Moreover, it challenges the conventional distinction between micrometastatic and macrometastatic disease, suggesting that the extent of nodal involvement is a continuum rather than a binary classification.

Detecting extracapsular infiltration poses a diagnostic challenge due to its subtle nature and the limitations of traditional imaging techniques like the above-mentioned US, CT, PET, and MRI. Particularly, US and MRI offer insights into nodal morphology and size but may fall short in reliably identifying extracapsular infiltration.

The presence of extracapsular infiltration warrants a multidisciplinary approach to treatment [17]. Exploring novel therapeutic strategies that specifically target the microenvironment and factors driving extracapsular invasion could potentially enhance treatment outcomes in this subgroup of patients.

## 2. Axillary Management

Therefore, for several years, the paradigm of axillary management in early clinically node-negative BC patients consisted of completion ALND if one metastatic ALN was detected with SLNB or AUS (and subsequently confirmed by histology) [18]. This approach changed when the Z0011 trial of the American College of Surgeons Oncology Group (ACOSOG) demonstrated in 2010 that women with small BC and up to two positive SLNs who did not proceed to ALND had similar recurrence-free survival as patients who underwent ALND [9].

Moreover, in 2013, the results of the IBCSG 23-01 trial were published, showing low incidence of axillary locoregional recurrences in patients with micro-metastases (detected by SLNB) whether they were treated with ALND or have not undergone any further axillary treatment, with further confirmation at long-term follow-up [19].

The results of these trials led to a radical change in the clinical practice, sparing completion ALND to many BC patients, thus reducing surgical complications with no adverse effect on overall survival [19].

Additionally, the impact of the status of the ALNs on prognosis is nowadays less important than in the past, since adjuvant treatment is increasingly tailored to BC biological features rather than the risk of local spread [10,20]. During the AMAROS trial, where patients with positive SLNB were randomly assigned to either ALND or axillary radiotherapy, the adjuvant treatment remained consistent in both groups. This indicates that detailed information on ALN status does not alter the indications for adjuvant treatment [21].

Therefore, after publication of such trials’ findings, assuming that local disease control can be achieved without ALND even in the presence of SLNs involvement and information on ALN status does not change either the type of adjuvant treatment or the prognosis, the role of SLNB in clinically node-negative early BC patients has been questioned, also considering its false negative rate which is estimated to be 8.4% (4–16%) [8,22], contrary to the low axillary recurrence rates, reported between 0 and 1.5% [23].

A new area of study and debate is the role of lymph node status or lymph node disease burden for adjuvant therapy planning. In HR-positive, HER2-negative patients with cN0 disease becoming pN0 or pN1 (1–3 positive lymph nodes), recent data from the TAILORx [24] trial suggests that SLNB staging may not be necessary for adjuvant therapy decision-making, emphasizing the value of genomic testing in BC management.

Moreover, a debated issue is represented by the management of patients in whom lymphoscintigraphy may fail to visualize the SLN. In the literature, reported rates of SLN non-visualization vary between 2 and 28%, depending on several clinical and tumor features [25]. In case of SLN’s unsuccessful visualization, the question of whether ALND should be performed in patients with early BC still represents a debated issue. Generally acceptable rules and recommendations on the need of ALND are lacking, both by scientific studies and international guidelines [25].

In our recent institutional retrospective study on 30,508 SLN procedures, we reported a rate of 1.7% of non-visualized SLNs, but with a high rate of intra-operative identification (73.3%), performing single tracer radioisotope SLNB [25]. Results revealed that lymph-nodes involvement was significantly associated with SLN non-identification during surgery (*p* < 0.001). In addition, we observed that 35% of patients with failed mapping and no sentinel node identified at time of surgery was pN0 on final staging, stressing the question of whether these patients could spare the burden of an ALND. Otherwise, Vereuhvel et al. underlined that ALND represents a recommendable choice for loco-regional control, according to the Dutch NABON guideline [26]. However, the variability in management of BC patients with SLN non-visualization was accentuated, given the role acquired by radiotherapy extended to the axilla and following the introduction of the criteria of the ACOSOG Z0011 trial, also looking to future findings of SOUND trial.

Even if the applicability of the Z0011 trial findings to non-visualized SLNs patients is uncertain, ALND does not appear as the preferred choice of breast surgeons in such conditions, compared to clinical-pathological characteristics, as highlighted in a Dutch Nationwide Survey study [26]. Furthermore, no statistically significant association was reported between ALND and improved survival in the absence of SLN visualization [26]. A scientific consensus for the management of this clinical condition of lymphoscintigraphic failure is therefore necessary to obtain specific recommendations useful in daily practice, further validating the role of defining the preoperative axillary clinical stage through an accurate radiological evaluation.

### 2.1. The ACOSOG Z0011 Trial

The ACOSOG Z0011 trial, also known as the Z0011 trial (American College of Surgeons Oncology Group Z0011 trial), is a landmark study in the field of BC management with important implications for the assessment and treatment of ALN in patients with BC. The trial was a randomized, controlled, and multicenter study that compared SLNB alone versus SLNB followed by ALND in early stage BC patients. Figure 1A.

Historically, it was a common practice to perform ALND for BC patients with positive SLNB in order to thoroughly assess the extent of ALN involvement and to potentially improve disease staging and control. However, this procedure could lead to complications such as lymphedema and shoulder dysfunction.

The trial, initiated in 1999 and completed in 2007, sought to determine if ALND was necessary for all patients with limited SLN involvement, focusing specifically on early stage BC patients (cT1-cT2, cN0) who were undergoing breast conserving surgery (BCS) with complementary whole-breast adjuvant radiation therapy. The primary objective was to compare outcomes between patients who underwent ALND and those who did not, specifically evaluating factors like disease recurrence, OS/DFS, and quality of life.

According to Z0011 trial, for women with early BC who received BCS and had no more than two involved SLNs, SLNB alone was found to be non-inferior to SLNB followed by ALND in terms of OS (86.3% vs. 83.6%, *p* = 0.02), DFS (80.2% vs. 78.2%, *p* = 0.32), Figure 1B,C, and loco-regional recurrence (1% vs. 0%). This held true as long as these patients also underwent standard whole breast irradiation and received adjuvant chemotherapy [7]. Additionally, the study revealed that SLNB alone was associated with a lower occurrence of lymphedema and other complications compared to SLNB followed by ALND. These results have led to a shift in the standard of care for ALN assessment in early stage BC, specifically in patients meeting specific criteria (size tumor < 5 cm, ALN involvement ≤ 2, and planned whole-breast radiation or systemic therapy), with SLNB alone now being considered the preferred approach in most cases. This has resulted in a significant reduction in the morbidity associated with ALND, while maintaining high levels of accuracy in disease staging [27].

The trial’s findings, published in 2011 in the Journal of the American Medical Association, led to a shift in clinical practice, with many experts advocating for more selective use of ALND in cases meeting the Z0011 trial’s criteria, supporting the “less is more” concept and emphasizing the importance of minimizing unnecessary surgical interventions to improve patient quality of life with reduction of potential complications.

Furthermore, in the ACOSOG Z0011 study, 23.7% of the ALND group patients had additional meta-static lymph nodes other than SLNs, without difference in the axillary recurrence rate or between the SLNB only and the ALND groups. Thus, these results increase questions about the need to re-move SLNs, especially in patients who undergo breast-conserving surgery and whole-breast radiation.

### 2.2. The IBCSG 23-01 Trial

ALND is not justified for patients with micrometastatic sentinel node involvement, and this does not impact on survival, as deduced by wide clinical trials [7]. A multicenter phase III trial (IBCSG 23-01) was published in 2013 in which 934 women with cT1 or cT2 tumors and micrometastatic disease (≤2 mm) in ≥1 SLN who underwent lumpectomy or mastectomy were randomized to receive ALND versus SLNB. Figure 2A After a median follow-up of over 5 years, no significant differences were observed in terms of axillary recurrence (1% in the observation group vs. <1% in the ALND group), and DFS (87.8% in the observation group vs. 84.4% in the ALND group, *p* = 0.16) between the two cohorts. Figure 2B,C However, the ALND group experienced significantly higher rates by about three times of severe complications, including sensory neuropathy (18% vs. 12%, *p* = 0.012), lymphedema (13% vs. 3%, *p* < 0.0001), and motor neuropathy (8% vs. 3%, *p* = 0.0004) [23]. Findings after a median follow-up of 9.7 years (IQR 7·8–12.7) consolidate those obtained at 5 years and are consistent with Z0011 trial analysis [19]. Conclusively, all these findings have supported the omission of ALND in cN0 early BC patients, irrespective of the type of breast surgery given (mastectomy or breast conserving surgery), in relation to a non-inferiority of observation approach versus ALND treatment, both in terms of DFS and regional recurrences, with potential reduction of morbidity. This demonstrates a clear trend of clinical research towards minimizing axillary surgery, even in the presence of SLN involvement. Opting for a therapeutic path that avoids ALND not only fosters a swifter post-operative recovery but also enhances the overall well-being of the patient, ensuring a quicker return to normal daily, work, and social. This consideration is pivotal in making the decision to embrace the optimal approach for each patient, considering the specific benefits and risks associated with the surgical approach.

### 2.3. The SOUND Trial

The SOUND (Sentinel node vs. Observation after axillary UltraSOuND) trial is a recent randomized controlled multicentric trial representing a further step forward in the conservative approach to BC aimed at improving patient’s quality of life [28]. Together with the ongoing INSEMA, BOOG 2013-08, SOAPET and NAUTILUS trials, it has compared SLNB to observation in clinically node negative patients [29,30,31].

It investigated the use of US to identify patients with early stage BC who can safely avoid SLNB, with the hypothesis of verifying the utility of the SLNB itself and the possible utility of AUS to safely identify a subgroup of patients with early stage BC who do not need SLNB.

The trial enrolled over 1000 women with clinically node-negative early BC (lesions ≤ 2 cm with negative pre-operative US or negative FNAC in case one doubtful node on US), randomized to either undergo SLNB (Group 1) or only observation (Group 2). All patients received conservative surgery followed by adjuvant radiotherapy and chemotherapy. In Group 1 patients, the standard clinical practice of SLNB and quadrantectomy at the same time was applied. If the sentinel node was negative, ALND was not performed. Considering recent findings, ALND was not performed even with minimal SLN involvement (SLN micro-metastases < 2 mm) [23]. In Group 2 patients, only quadrantectomy was performed without SLNB. The primary endpoint was the rate of ALN metastasis in the observation arm and their distant disease-free survival; other secondary endpoints were quality of life and evaluation of type of adjuvant treatment administered.

The SOUND trial was closed in 2017- and 5-years outcome results analysis are ongoing.

The reported sensitivity and specificity of AUS ranges from 26.4% to 94% and 53% to 98.1%, respectively [32,33]. Therefore, results of the SOUND trial might have important implications for the management of early stage BC, as they could provide evidence to support the use of AUS as a tool to avoid unnecessary SLNB in a subset of patients. This can help to reduce the morbidity associated with SLNB, such as lymphedema, seroma, bleeding, and neurological injuries [34,35].

## 3. Post-Neoadjuvant Chemotherapy

Current prospective randomized surgical trials are actively exploring the potential de-escalation of axillary surgery in the neoadjuvant setting. Specifically, they are investigating the role of SLNB for certain subgroups of BC patients who have been treated with neoadjuvant therapy (NT) and show clinical complete response (cN0). The decision to omit SLNB in these cases is a subject of ongoing debate and study. Two European trials, EUBREAST-01 and ASICS, are currently underway as single-arm studies. These trials focus on patients with the highest probability of achieving a pathological complete response after NT, particularly those with triple-negative or HER2-positive subtypes. The evaluation of radiological complete response post-NT by imaging is being used as the basis for testing the omission of SLNB [36,37]. The outcomes of these ongoing trials could provide valuable insights to surgeons in determining the appropriate approach to breast surgery and the extent of axillary treatment based primarily on the response to NT, as determined by specific imaging assessments.

## 4. Emerging Techniques

Several emerging techniques are being investigated for ALN assessment in BC, including magnetic resonance lymphography (MRL), superparamagnetic iron oxide-enhanced magnetic resonance imaging (SPIO-MRI), diffusion-weighted imaging (DWI), and molecular imaging [38].

MRL is a non-invasive technique that uses injection of gadolinium contrast and MRI to visualize lymph nodes. This procedure involves injecting contrast into the lymph nodes in the groin and tracking its flow through the lymphatic system using T1-weighted MR images. At present, this technique has demonstrated successful application in imaging and planning the treatment of conditions affecting the thoracic duct, lymphatic leaks, and other lymphatic abnormalities, including plastic bronchitis [39].

The use of SPIO-MRI has been suggested as a non-ionizing radiation method for visualizing malignant lymph nodes. This technique is capable of detecting small nodes that harbor metastases [40]. SPIOs are taken up naturally by macrophages, leading to a low-signal region in normal lymph node areas. Areas within the node that do not show this signal loss are likely involved with a tumor, although other factors like fibrosis or inflammation can also cause similar results.

The effectiveness of SPIO-MRI in detecting metastases in SLNs of BC patients has been evaluated [41]. A lymph node with a high-signal-intensity area on T2-weighted MRI, whether throughout the node or in a focal region, is considered to be involved with metastasis. The sensitivity and specificity of MRI for detecting SLN metastases were found to be 84% and 91%, respectively. Based on SPIO-MRI results, SLNB could potentially be avoided in BC patients with non-metastatic SLNs [41]. However, the lack of clinically available and approved USPIOs (ultra-small superparamagnetic iron oxides) currently hinders broader adoption and larger studies.

DWI is a functional MRI technique that uses the diffusion of water molecules in tissue to provide information about cellularity and tissue architecture [42]. The inclusion of functional MRI, specifically using diffusion-weighted MRI, has demonstrated superiority over the conventional MRI protocol in assessing lymph node status, both qualitatively and quantitatively [43].

Molecular imaging techniques, such as positron emission mammography (PEM) and molecular breast imaging, use radiotracers to detect molecular targets associated with cancer cells, such as glucose metabolism and estrogen receptors [44].

## 5. Artificial Intelligence and ALN Status

Artificial Intelligence (AI), particularly deep learning (DL) algorithms using convolutional neural networks (CNNs), has gained popularity in the medical community for revolutionizing the diagnosis of diseases based on image analysis [45,46]. CNNs are a specific type of DL algorithm commonly used to analyze BC images, as they excel in determining image features [47]. These emerging innovations have shown to enhance the accuracy of US and MRI in assessing ALNs by providing automated image segmentation and radiomics feature extraction.

A recent study [48] explored an AI system implemented through Google Cloud AutoML Vision, which classifies preoperative ALNs of BC patients as benign or malignant. The system’s results were compared to blind readings from three experienced radiologists. The AI performed similarly to the trained radiologists, showing slightly less sensitivity but greater specificity in external validation. Although the differences between the AI and radiologist groups were not statistically significant, the study concluded that combining the AI system with radiologists in practice could optimize results.

In another investigation, Guo et al. identified 937 BC patients with prior US images to train and test two different DL radiomics models for evaluating SLN and non-SLN metastasis. The DL radiomics models demonstrated a sensitivity of 89.7% for predicting the risk of SLN metastasis and 98.4% for non-SLN metastasis [49]. These larger-scale studies indicate that DL models evaluating US images show promise as an early diagnostic tool for assessing lymph node status in primary BC patients.

Furthermore, the capabilities of AI models can be further improved by integrating multiple US modalities. Zheng et al. combined DL radiomics of conventional US with shear wave elastography to predict ALN metastasis pre-operatively [50]. Shear wave elastography provides information about the elasticity of the tissue, particularly the stiffness of the tumor. A higher shear wave velocity has been correlated with a higher probability of metastasis, and when combined with DL radiomics, the model accurately predicts the metastatic status of ALNs and can differentiate between ALNs with low versus high metastatic burden.

MRI is another imaging modality utilized in diagnosing ALN metastasis in BC patients. Ren et al. developed a CNN approach to identify ALN metastasis in breast MRI images, which performed better than an experienced radiologist [51]. Similarly, radiomic signatures in dynamic contrast-enhanced MRI were examined to create a novel clinical-radiomic nomogram that accurately identifies ALN metastasis in early BC patients [52].

Overall, DL models for predicting BC metastasis to ALNs before surgery have shown early success, with higher sensitivity and specificity compared to experienced radiologists. However, further work is needed, with larger-scale studies and standardized variables, to establish their clinical usefulness and interpretability.

## 6. Challenges and Opportunities

Despite the advancements in ALN assessment techniques, there are still several challenges that need to be addressed. One of the main challenges is the lack of standardization in imaging protocols and interpretation criteria. Another challenge is the different radiologists’ level of experience and the variability in the sensitivity and specificity of different techniques, depending on factors such as tumor characteristics and nodal involvement. Additionally, emerging techniques require further validation in large-scale clinical trials to evaluate their efficacy and cost-effectiveness. Future research should focus on developing standardized protocols and interpretation criteria for ALN assessment, as well as investigating the efficacy of emerging techniques in large-scale clinical trials.

## 7. Conclusions

The assessment of ALNs in BC patients has undergone transformative advancements that hold great promise for improving patient care while minimizing morbidity. The understanding that ALN involvement significantly impacts prognosis and treatment decisions has driven the development of more accurate and less invasive assessment techniques. Transition from routine ALND to SLNB, as demonstrated by landmark trials like Z0011 and IBCSG 23-01, has revolutionized the field by reducing postoperative complications while maintaining effective disease management.

The emergence of innovative imaging modalities, such as US, MRI, and molecular imaging, presents an opportunity to enhance ALN evaluation further. Integration of AI algorithms into image analysis has demonstrated the potential to significantly augment diagnostic accuracy and provide more personalized treatment strategies. These approaches offer a glimpse into the future of precise and tailored BC management.

Despite these advancements, challenges remain in terms of standardizing imaging protocols, ensuring consistent interpretation, and conducting large-scale clinical validations of emerging techniques. Ongoing research, exemplified by initiatives like the SOUND trial, reflects the dedication of the medical community to refine ALN assessment strategies. These trials also highlight the evolving role of AI in facilitating informed decision-making by assisting medical professionals in diagnosing and planning treatments.

As the landscape of breast cancer management evolves, the goal is clear: to strike a balance between accurate disease evaluation and minimizing invasiveness. Collaboration among radiologists, oncologists, and researchers is paramount to refine and implement these new approaches effectively. By leveraging the power of cutting-edge imaging and AI-driven solutions, medical practitioners are paving the way for a future where BC patients can benefit from both improved clinical outcomes and an enhanced quality of life. The continuous exploration and integration of advanced technologies will undoubtedly shape the future of ALN assessment, empowering medical professionals to offer more personalized and effective care to BC patients.

## Figures and Tables

**Figure 1 medicina-59-01544-f001:**
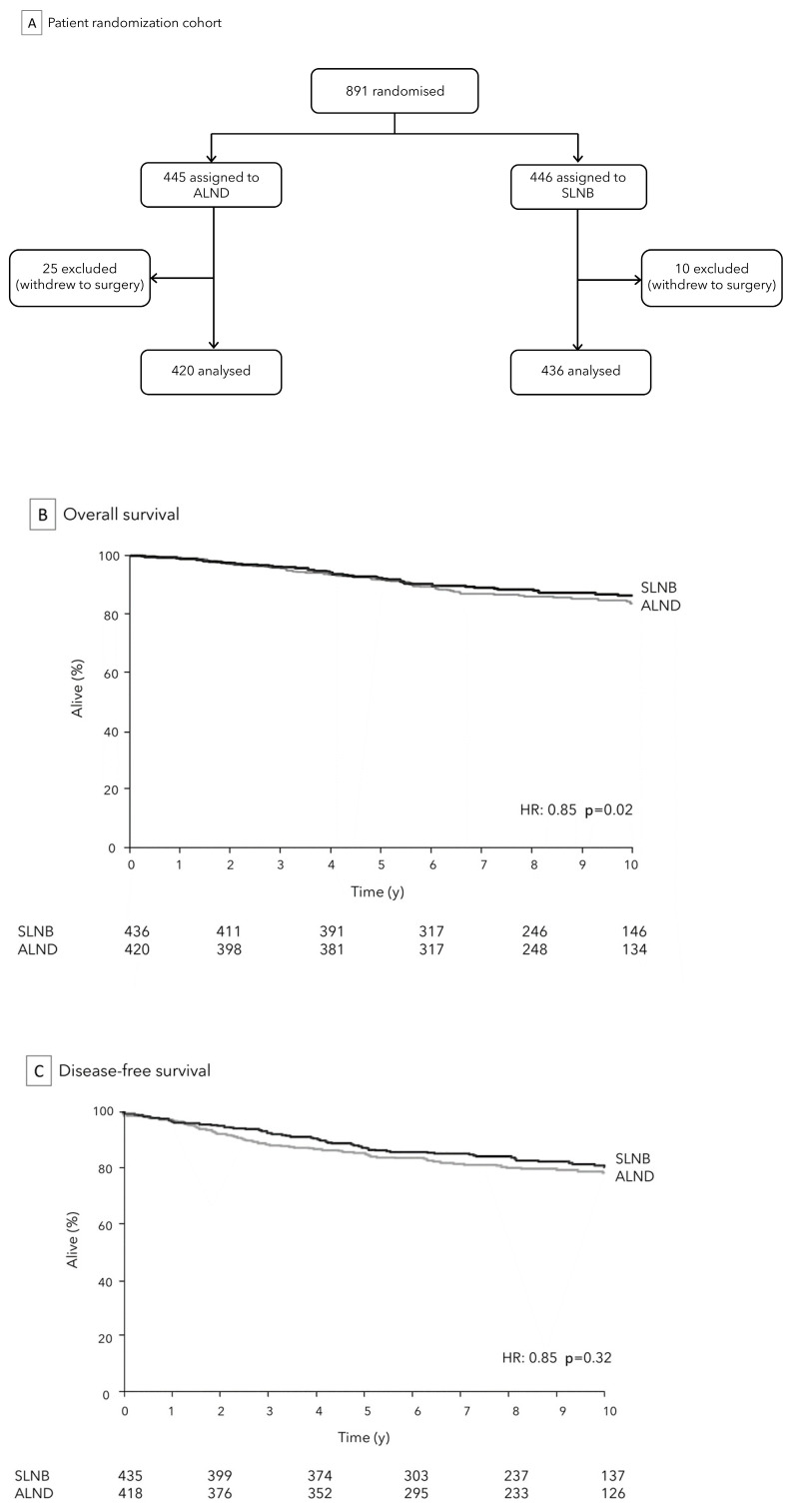
ACOSOG Z0011 Trial: analytic cohort (**A**) and comparison of SLNB to ALND for OS (**B**) and DFS (**C**).

**Figure 2 medicina-59-01544-f002:**
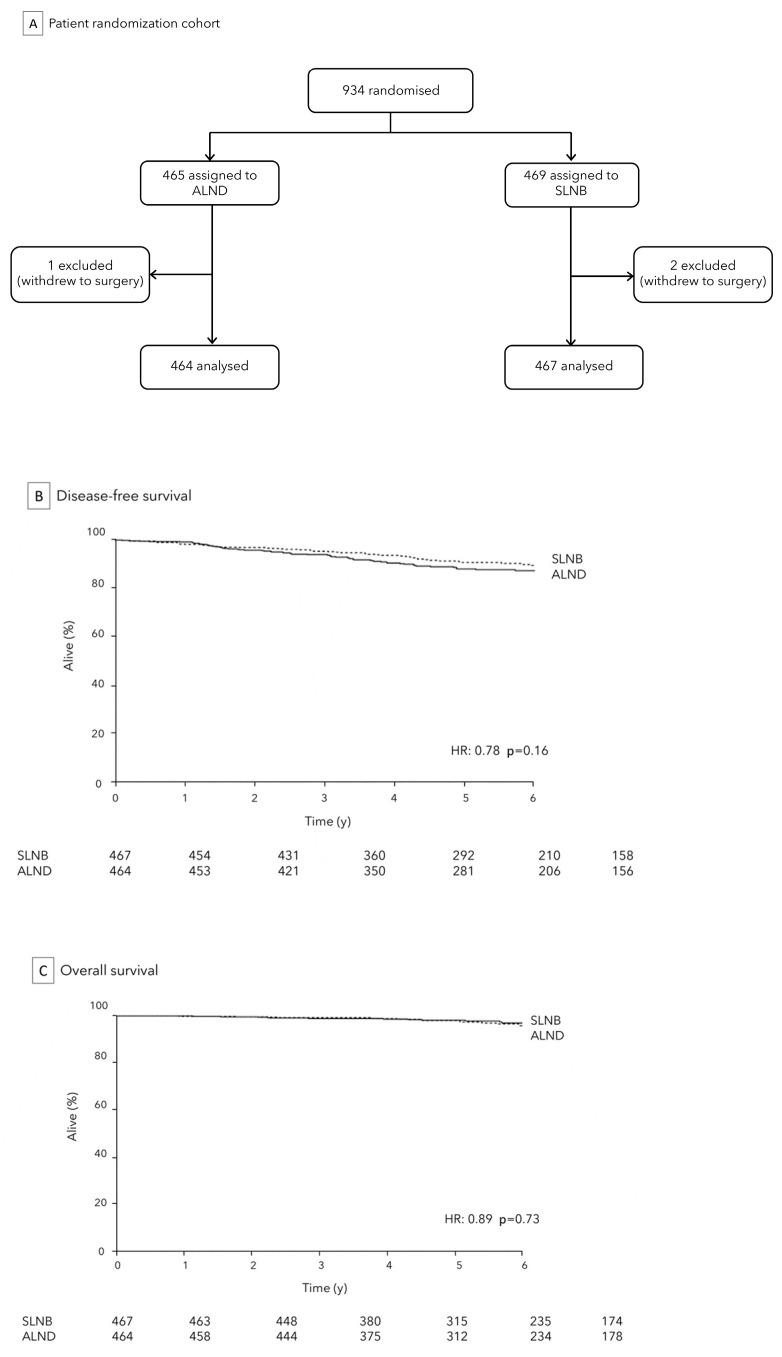
IBCSG 23-01 Trial: analytic cohort (**A**) and comparison of SLNB to ALND for DFS (**B**) and OS (**C**).

## Data Availability

The data presented in this study are available upon request from the corresponding author. The data are not publicly available due to restrictions.

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
