# Peer review of "Future Directions in the Assessment of Axillary Lymph Nodes in Patients with Breast Cancer"

_medicina, 2023, doi:10.3390/medicina59091544_

Round 1

Reviewer 1 Report

Overall, a clearly written manuscript summarizing the most important studies concerning surgical interventions in the axilla for breast cancer. The studies cited at the beginning have been known for many years, so this part of the manuscript might be shorter, especially if breast surgeons are to be the main reader group. Conversely, the second part, which refers to ongoing studies and highlights future prospects, could be somewhat more extensive.

As it stands, the manuscript will be of interest to readers with little prior knowledge of breast surgery.

Author Response

Dear Reviewer,

Thank you for your valuable feedback on our manuscript. We genuinely appreciate your time and insights.

We are pleased to hear that the overall clarity of the manuscript has been positively received. Your suggestion to potentially shorten the section discussing well-known studies at the beginning of the manuscript is well taken. We agree that for our intended audience, especially breast surgeons, focusing on the most pertinent and recent information is crucial. Accordingly, we have revisited and streamlined this section to ensure that it provides the necessary context without unnecessary length, as you can find find in the attached version of our reviewed paper.

Particularly, the following sentences in the paragraph “Current techniques” in the text has been removed:

-“In the past, ALND, namely the levels I and II surgical excision, was accepted as the standard treatment [3; 4]. Level III dis-section remains controversial and discussed in specific clinical high-risk conditions, especially in the presence of gross axillary disease [5].”

-“Thanks to results from historical wide trials [9;10], SLNB in the clinically node negative patients was demonstrated to minimize morbidity without affecting oncologic out-comes. SLNB is performed also in selected patients with ductal carcinoma in situ (DCIS) in whom there is a substantial risk of an upgrade of the lesion at final pathology [7], i.e. a considerable mass indicating a high likelihood of invasive cancer on breast imaging and physical examination, or a substantial area of 5 cm or more of ductal carcinoma in situ (DCIS) observed in imaging, or when mastectomy is deemed necessary, in line with the confirmed recommendations from the 2014 ASCO update [11].”

Furthermore, your recommendation to expand the second part of the manuscript, which discusses ongoing studies and future prospects, is well noted. We understand the significance of providing in-depth insights into current research directions and potential advancements: in fact, we had included the paragraph relating to the new studies in progress regarding the role of diagnostics in the axillary diagnostic assessment in the neoadjuvant setting; in the revised version we have however added a mention to TAILORx and RxPONDER trials, which represents new perspectives regarding the role of the nodal disease burden on decision making process for adjuvant treatments. We have not included further mentions of ongoing or recently closed studies relating to the conservative evolution of axillary surgery, such as POSNOC or SENOMAC trials, given that the paper is not centered essentially on surgical issues, as rightly pointed out in your comments. However, we have analyzed the SOUND study in detail with specific reference to further ongoing studies examining SLNB vs. axillary observation in patients with normal axillary imaging (INSEMA, BOOG 2013-08, SOAPET and NAUTILUS trials). We made changes in our reviewed manuscript accordingly adding the following section in the paragraph “Axillary management”.

A new area of study and debate is the role of lymph node status or lymph node disease burden for adjuvant therapy planning. In HR-positive, HER2-negative patients with cN0 disease becoming pN0 or pN1 (1-3 positive lymph nodes), recent data from the TAILORx trial suggests that SLNB staging may not be necessary for adjuvant therapy decision-making, emphasizing the value of genomic testing in BC management.”

Your assessment that the manuscript will be valuable to readers with limited prior knowledge of breast surgery is reassuring. We aim to make our work accessible and informative for a wide range of readers, and your feedback validates that we are on the right track.

We hope that this new version of our paper is now worthy of publication in the MEDICINA Journal. If you have any further suggestions or concerns, we would be grateful to receive them.

Once again, thank you for your thoughtful comments and constructive suggestions: we are committed to improving the manuscript based on your input and creating a more impactful contribution to the field.

Kind regards,

The Authors

Reviewer 2 Report

I would like to congratulate the authors of the article on the work done.

The aim of this study was to summarize the current state of assessment of axillary lymph nodes in breast cancer patients and provide insight into future directions.

The presented problem is not a new issue, but very important for clinicians. The authors present all findings in a convincing way, especially the practical value and clinical benefits resulting from the aim stated in their article.

However, two important issues are missing. The authors did not present current recommendations for the management in the absence of visualization of the sentinel lymph node. I also did not find a discussion of the problem of the presence of extracapsular infiltration of metastatic lesions in the sentinel node. This should be supplemented based on current literature.

In terms of formal requirements, the concept of the article is compliant. The authors have employed a method which complies with the criteria to be met by scientific papers.

Author Response

Dear Reviewer,

Thank you very much for your insightful and encouraging feedback on our article. We truly appreciate your kind words and your recognition of the efforts we put into the research.

Your observations are invaluable, and we are grateful for your thoughtful suggestions. Your point regarding the absence of current recommendations for managing cases where visualization of the sentinel lymph node is not achieved is well taken. We understand the importance of addressing this aspect comprehensively, and we are included the following section in the article that discusses the management strategies in such scenarios.

The considerations about the issue of lack of SLN visualization have been added at the end of “Axillary management” paragraph, as follows:

 “Moreover, a debated issue is represented by the management of patients in whom lymphoscintigraphy may fail to visualize the SLN. In the literature, reported rates of SLN nonvisualization vary between 2 and 28%, depending on several clinical and tumor features [Magnoni F, Corso G, Gilardi L, et al. Does failed mapping predict sentinel lymph node metastasis in cN0 breast cancer? Future Oncol. 2022;18(2):193-204. doi:10.2217/fon-2021-0470]. In case of SLN’s unsuccessful visualization, the question if ALND should be performed in patients with early BC still represent a debated issue. Generally acceptable rules and recommendations on the need of ALND are lacking, both by scientific studies and international guidelines [Magnoni F, Corso G, Gilardi L, et al. Does failed mapping predict sentinel lymph node metastasis in cN0 breast cancer? Future Oncol. 2022;18(2):193-204. doi:10.2217/fon-2021-0470].

In our recent institutional retrospective study on 30,508 SLN procedures, we reported a rate of 1.7% of non-visualized SLNs, but with a high rate of intra-operative identification (73.3%), performing single tracer radioisotope SLNB [Magnoni F, Corso G, Gilardi L, et al. Does failed mapping predict sentinel lymph node metastasis in cN0 breast cancer? Future Oncol. 2022;18(2):193-204. doi:10.2217/fon-2021-0470]. Results revealed that lymph-nodes involvement was significantly associated with SLN non-identification during surgery (P<0.001). In addition, we observed that 35% of patients with failed mapping and no sentinel node identified at time of surgery was pN0 on final staging, stressing the question if these patients could spare the burden of an ALND.  Otherwise, Vereuhvel and colleagues underlined that ALND represents a recommendable choice for loco-regional control, according to the Dutch NABON guideline [Verheuvel NC, Voogd AC, Tjan-Heijnen VCG, Roumen RMH. What to Do with Non-visualized Sentinel Nodes? A Dutch Nationwide Survey Study. Ann Surg Oncol. 24(8), 2155-2160 (2017)].

However, the variability in management of BC patients with SLN non-visualization was certainly accentuated given the role acquired by radiotherapy extended to the axilla [Lyman GH, Giuliano AE, Somerfield MR et al. American Society of Clinical Oncology guideline recommendations for sentinel lymph node biopsy in early-stage breast cancer. J Clin Oncol. 23(30), 7703-7720 (2015)], and following the introduction of the criteria of the ACOSOG Z0011 trial, also looking to future findings of SOUND trial.

Even if applicability of Z0011 trial findings to non-visualized SLNs patients is uncertain, ALND does not appear as the preferred choice of breast surgeons in such conditions, compared to clinical-pathological characteristics, as highlighted in a Dutch Nationwide Survey study [Verheuvel NC, Voogd AC, Tjan-Heijnen VCG, Roumen RMH. What to Do with Non-visualized Sentinel Nodes? A Dutch Nationwide Survey Study. Ann Surg Oncol. 24(8), 2155-2160 (2017)].

Furthermore, no statistically significant association was reported between ALND and improved survival in the absence of SLN visualization [Verheuvel NC, Voogd AC, Tjan-Heijnen VCG, Siesling S, Roumen RMH. Non-visualized sentinel nodes in breast cancer patients; prevalence, risk factors, and prognosis. Breast Cancer Res Treat. 167(1), 147-156 (2018)]. A scientific consensus for the management of this clinical condition of lymphoscintigraphic failure is therefore necessary to obtain specific recommendations useful in daily practice, further validating the role of defining the preoperative axillary clinical stage through an accurate radiological evaluation.

This addition will undoubtedly enhance the practical applicability of our work and provide clinicians with more comprehensive insights.

Additionally, we acknowledge your recommendation to include a discussion on the topic of extracapsular infiltration of metastatic lesions in the sentinel node. This is indeed a significant consideration. Accordingly, we included the following section into the reviewed version of our manuscript:

Extracapsular infiltration refers to the extension of tumor cells beyond the lymph node capsule into the surrounding perinodal tissue. This phenomenon signifies a more ad-vanced stage of nodal involvement and is associated with increased tumor burden, higher rates of recurrence, and reduced overall survival rates ]19]. Fujii et al. [19] performed a bi-opsy of the SLN on 276 patients diagnosed with BC who had clinically negative ALNs and found that all cases of positive nodes in non-SLN had extracapsular infiltration at the metastatic SLNs. Their results suggested that the presence of extracapsular infiltration at metastatic SLNs is a strong predictor for residual disease in the axilla. Moreover, it chal-lenges the conventional distinction between micrometastatic and macrometastatic disease, suggesting that the extent of nodal involvement is a continuum rather than a binary clas-sification.

Detecting extracapsular infiltration poses a diagnostic challenge due to its subtle nature and the limitations of traditional imaging techniques like the above-mentioned US, CT, PET and MRI. Particularly, US and MRI offer insights into nodal morphology and size but may fall short in reliably identifying extracapsular infiltration.

The presence of extracapsular infiltration warrants a multidisciplinary approach to treat-ment [19]. Exploring novel therapeutic strategies that specifically target the microenviron-ment and factors driving extracapsular invasion could potentially enhance treatment out-comes in this subgroup of patients.

Your guidance helps us improve the quality and relevance of our work, and we are genuinely grateful for your time and effort in reviewing our article. We hope that this new version of our paper is now worthy of publication in the MEDICINA Journal. If you have any further suggestions or concerns, we would be grateful to receive them.

Once again, thank you for your kind words and constructive feedback.

Kind regards,

The Authors

Reviewer 3 Report

Future Directions in the Assessment of Axillary Lymph Nodes in Patients with Breast Cancer.

The review also examines landmark clinical trials that have influenced axillary management, such as the Z0011 trial and the IBCSG 23-01 trial. The role of artificial intelligence, specifically deep learning algorithms, in improving ALN assessment is examined. Results: The review outlines the key findings of these trials, which demonstrated the feasibility of avoiding axillary lymph node dissection in certain patient populations with low sentinel lymph node burden. It also discusses ongoing trials, including the SOUND trial, which investigates the use of axillary ultrasound to identify patients who can safely avoid sentinel lymph node biopsy.

The authors should expand the discussions of the two clinical trials.

Tables and figures must be included.

The conclusion section must be expanded.

Author Response

Dear Reviewer,

We would like to express our sincere gratitude for your careful and insightful review of our manuscript. Your feedback has been invaluable in refining our work to meet the highest standards of academic quality. We have carefully considered your comments and suggestions, and we are pleased to share the following revisions we have made to address your concerns:

  1. We appreciate your suggestion to expand the discussions on the landmark clinical trials that have influenced axillary management, namely the Z0011 trial and the IBCSG 23-01 trial. In response, we have enriched our manuscript by providing more comprehensive insights into these trials, including a detailed exploration of their methodologies, key findings, and implications. This expansion ensures that readers gain a deeper understanding of the pivotal role these trials have played in shaping axillary management strategies.
  2. Such expansion includes also the addition of Tables and Figures as you rightly suggested: we acknowledge the importance of visual aids in enhancing the clarity and presentation of complex information indeed. In line with your recommendation, we have incorporated relevant graphs and tables combined in figures 1 and 2 focusing on the IBCSG 23-01 and Z0011 trials, respectively. These visual elements effectively illustrate the data discussed in the text, thereby enhancing the overall readability and impact of our work.
  3. Recognizing the need for a comprehensive conclusion that encapsulates the key insights of our review, we have expanded the conclusion section as follows:

The assessment of ALNs in BC patients has undergone transformative advancements that hold great promise for improving patient care while minimizing morbidity. The understanding that ALN involvement significantly impacts prognosis and treatment decisions has driven the development of more accurate and less invasive assessment techniques. Transition from routine ALND to SLNB, as demonstrated by landmark trials like Z0011 and IBCSG 23-01, has revolutionized the field by reducing postoperative complications while maintaining effective disease management.

Emergence of innovative imaging modalities, such as US, MRI, and molecular imaging, presents an opportunity to enhance ALN evaluation further. Integration of AI algorithms into image analysis has demonstrated the potential to significantly augment diagnostic accuracy and provide more personalized treatment strategies. These approaches offer a glimpse into the future of precise and tailored BC management.

Despite these advancements, challenges remain in terms of standardizing imaging protocols, ensuring consistent interpretation, and conducting large-scale clinical validations of emerging techniques. Ongoing research, exemplified by initiatives like the SOUND trial, reflects the dedication of the medical community to refine ALN assessment strategies. These trials also highlight the evolving role of AI in facilitating informed decision-making by assisting medical professionals in diagnosing and planning treatments.

As the landscape of breast cancer management evolves, the goal is clear: to strike a balance between accurate disease evaluation and minimizing invasiveness. Collaboration among radiologists, oncologists, and researchers is paramount to refine and implement these new approaches effectively. By leveraging the power of cutting-edge imaging and AI-driven solutions, medical practitioners are paving the way for a future where BC patients can benefit from both improved clinical outcomes and an enhanced quality of life. The continuous exploration and integration of advanced technologies will undoubtedly shape the future of ALN assessment, empowering medical professionals to offer more personalized and effective care to BC patients.

This expansion allows us to succinctly summarize the main takeaways of the manuscript, highlighting the implications of our findings for future axillary lymph node assessment strategies and patient care.

We are confident that these revisions significantly enhance the quality and comprehensiveness of our manuscript. We are grateful for your meticulous assessment, which has undoubtedly contributed to the refinement of our work. Your feedback has been instrumental in shaping the manuscript into a more impactful contribution to the field.

We hope that this new version of our paper is now worthy of publication in the MEDICINA Journal. If you have any further suggestions or concerns, we would be grateful to receive them. We value your expertise and are committed to making necessary revisions to enhance the quality of our work.

Thank you again for your valuable input.

Sincerely,

The authors

Round 2

Reviewer 3 Report

Future Directions in the Assessment of Axillary Lymph Nodes in Patients with Breast Cancer

 The review outlines the key findings of these trials, which demonstrated the feasibility of avoiding axillary lymph node dissection in certain patient populations with low sentinel lymph node burden. It also discusses ongoing trials, including the SOUND trial, which investigates the use of axillary ultrasound to identify patients who can safely avoid sentinel lymph node biopsy.

The authors have included all the changes suggested by the reviewer. The manuscript does not require any other changes.